# Investigating the Antibacterial Properties of Prospective Scabicides

**DOI:** 10.3390/biomedicines10123287

**Published:** 2022-12-19

**Authors:** Sara Taylor, Deonne Walther, Deepani D. Fernando, Pearl Swe-Kay, Katja Fischer

**Affiliations:** Infection and Inflammation Program, Scabies Laboratory, QIMR Berghofer MRI, Brisbane, QLD 4006, Australia

**Keywords:** scabies, impetigo, mānuka oil, abametapir, ivermectin, β-triketones, *Staphylococcus aureus*, *Streptococcus pyogenes*, *Streptococcus dysgalactiae* subsp. *equisimilis*, *Acinetobacter baumannii*

## Abstract

Scabies is a dermatological disease found worldwide. Mainly in tropical regions, it is also the cause of significant morbidity and mortality due to its association with potentially severe secondary bacterial infections. Current treatment strategies for scabies do not consider the role of opportunistic bacteria, and here we investigate whether current and emerging scabicides can offer any anti-bacterial protection. Using the broth microdilution method, we examined antimicrobial potential of the current scabicide ivermectin and emerging scabies treatments: abametapir, mānuka oil, and its individual β-triketones. Our results demonstrate that the two novel scabicides abametapir and mānuka oil have antimicrobial properties against common scabies-associated bacteria, specifically *Staphylococcus aureus*, *Streptococcus pyogenes*, *Streptococcus dysgalactiae* subsp. *equisimilis* and *Acinetobacter baumannii*. The current scabicide ivermectin offers some antimicrobial activity and is capable of inhibiting the growth aforementioned bacteria. This research is important as it could help to inform future best treatment options of scabies, and scabies-related impetigo.

## 1. Introduction

Scabies is a highly contagious and pruritic skin disease caused by the obligate parasitic mite *Sarcoptes scabiei*, and is amongst the most common dermatological skin diseases worldwide with an estimated global prevalence of ~400 million cases annually [1]. Scabies is particularly prevalent in resource-poor populations where the disease is often endemic due to poverty and overcrowded living conditions, insufficient health care and a normalization of skin conditions along with other common childhood illnesses. In 2017, scabies was recognized as a neglected tropical disease by the World Health Organisation (WHO), a classification that came with calls for increased research into novel drugs and diagnostics [2]. Currently, scabies treatment varies significantly world-wide and concurrent treatment of associated secondary bacterial infections is often not considered. The two primary drugs in use are broad-spectrum anti-parasiticides, namely ivermectin (oral) and permethrin (topical). Both of these drugs are neuroinhibitors and target only the motile stages of the parasite’s lifecycle, which necessitates repeat treatments [3]. Consistent with other single-target agents, variable scabicidal efficacy and drug tolerability by the parasite have been observed for these drugs in recent years [4], clinical resistance to ivermectin and anecdotal evidence of resistance to permethrin have been reported [5,6] and the link to severe downstream complications due to bacterial co-infections has become more apparent [7,8,9,10]. These issues have renewed the focus on the development of new-generation scabicides. In our laboratory, two emerging parasiticides are being investigated for scabicidal activity: abametapir, an Australian new-generation lousicide and mānuka oil, an essential oil derived from the plant *Leptospermum scoparium*, native to Australasia.

### 1.1. Emerging Scabicides

Despite the clear and urgent need for new treatment options, very few new or repurposed scabicides have been reported in the last 30 years. As we are dealing with a neglected tropical disease, a significant challenge lies in the requirement for drugs that are both readily accessible and cheaply produced. One approach that has shown some promise is the repurposing of drugs developed against other obligate ectoparasites. Abametapir (Figure 1) is a metalloproteinase inhibitor that is effective as a single dose treatment against *Pediculus humanus capitis* (head lice), due to its ability to kill both the motile and the egg stages of this parasite [11]. It is believed to chelate heavy metal cations, thereby inhibiting metalloproteinases, a group of catalytic enzymes essential in all life stages of the head lice. Metalloproteinases have been shown to be particularly important for the development of the louse ova and for egg hatching [12,13]. In a randomized double-blind study a 0.74% abametapir lotion was shown to completely inhibit egg hatching [14], and for this reason it has been considered as a potential new-generation scabicide. Like scabies mites, head lice are obligate ectoparasites in humans, and similarly to current scabies treatments, previous head lice treatments have relied on neurotoxic properties, thereby only targeting the motile stages of the parasitic lifecycle, and necessitating repeat treatment [14]. Increased drug resistance was observed in head lice to neurotoxic lousicides [15,16], and the introduction of abametapir as a multi-target lousicide in 2020 may indeed be crucial for the control of this disease in the future. Metalloproteases are abundant in all stages of the scabies mite lifecycle, and therefore abametapir is a good candidate scabicide [17]. The proposed mechanism of action of abametapir and the ubiquitous nature of metalloproteases in all bacterial species indicate that abametapir may also have unexplored antibacterial properties.

Plant essential oils (EOs), are being explored as safe and widely acceptable natural treatment options for many neglected tropical skin diseases. In particular, extracts from *L. scoparium* (mānuka, from the *Myrtaceae* family) have been found to be effective against a number of arthropods [18,19,20,21], including promising effects against *S. scabiei* [22]. Mānuka oil has been used traditionally as a natural antimicrobial to treat wounds and skin infections [23]. Potent broad-spectrum activity has been demonstrated against Gram-positive bacteria, including strains of methicillin-resistant *S. aureus* (MRSA), some Streptococcus species, and several species of pathogenic fungi [24,25,26,27,28]. Chemically, mānuka oil is a complex mixture comprised mainly of terpenes, terpenoids, and β-triketones, the latter of which have been proposed to be associated with acaricidal [19] and antimicrobial [19,23,29,30] activity observed in *Myrtaceae* EOs and extracts. In mānuka oil, the major β-triketones present are flavesone, leptospermone, isoleptospermone and to a lesser extent, grandiflorone (Figure 1). While the antimicrobial properties of mānuka oil are well-established in the literature [27], there are few studies addressing the antimicrobial activity of the individual β-triketones [24].

### 1.2. The Importance of Anti-Bacterial Activity in Addition to Acaricidal Properties in a Scabies Treatment

Scabies is strongly associated with secondary bacterial infections due to a combination of factors, including the burrowing of the mites, the release of allergens causing itch and the epithelial damage from intense scratching [31,32]. Additionally, it is known that *S. scabiei* mites release immune suppressive molecule via their feces and saliva, which inhibit the hosts complement system [31,32,33], creating an ideal microenvironment for the development of opportunistic bacterial infections [31,32,33]. These factors contribute to the substantial burden of bacterial diseases associated with scabies, particularly in tropical climates. Notably, the superinfections in scabies appear to be uncommon in European countries [34]. Based primarily on epidemiological data, the two main pathogens associated with scabies are *S. aureus* and *Streptococcus pyogenes* (Group A Streptococcus/GAS). These two pathogens are linked to cellulitis, bacteremia, post-streptococcal glomerulonephritis (PSGN), acute rheumatic fever (ARF) and rheumatic heart disease (RHD) [9,10]. We have also reported the presence of another two medically important pathogens; *Streptococcus dysgalactiae* subsp. *equisimilis* (Group G Streptococcus/GGS) and *Acinetobacter baumanii* in scabies infected human skin [8]. It was proposed that when the epidermal barrier and normal microbial composition are disrupted due to scabies infestation, the burrowing scabies mites release molecules into the skin that interfere with the host innate immune response, promoting the growth of opportunistic bacteria, which in turn may switch on their virulent traits [35,36].

Currently, little is known about the relationships between scabies parasites and scabies-associated bacterial pathogens, even though the latter are the major cause of scabies morbidity and mortality [7]. There is, however, an urgent need to understand how anti-scabies treatment impacts on pathogens associated with the mite. To establish the therapeutic potential of emerging scabicides with regards to preventing secondary infections, we investigated their activity against pathogens commonly associated with scabies related impetigo, including *S. aureus*, *S. pyogenes*, *S. dysgalactiae*, and *A. baumannii*. The activity of a commercial mānuka oil (with high β-triketones concentration), synthesized β-triketones (flavesone, leptospermone, isoleptospermone, and grandiflorone), and abametapir were compared to that of pure ivermectin or commercial topical formulation of ivermectin (Ivomec).

## 2. Materials and Methods

### 2.1. Synthesis

Both 1D and 2D nuclear magnetic resonance (NMR) spectra were acquired on a Bruker Avance 300 MHz or 500 MHz spectrometer at 298 K. All ^13^C NMR spectra were recorded at 125 MHz on a Bruker Avance. Coupling constants are given in Hertz (Hz) and chemical shifts are reported as δ values in parts-per-million (ppm), with the solvent resonance as the internal standard (^1^H NMR-CDCl_3_: δ 7.26 and ^13^C NMR-CDCl_3_: δ 77.0; ^1^H NMR-(CD_3_)_2_CO: δ 2.05 and ^13^C NMR-(CD_3_)_2_CO: δ 29.9; Appendix A). Samples for GC-MS experiments were dissolved in reagent grade acetone (Merck) to a concentration of ~1 mg/mL. GC-MS experiments were performed using a Shimadzu QP2020 NX GC-MS equipped with an Rxi-5ms column (30 m, 0.25 mm ID, 0.25 μm) and using He carrier gas. Reactions were monitored by GC-MS (as described above) or thin layer chromatography (TLC) using silica gel 60 F254 TLC plates.

#### 2.1.1. General Synthesis of Acylphloroglucinols

Synthesis of acylated phloroglucinols was performed according to a modified method from those reported by George et al. [37]. Phloroglucinol (1.00 eq) was dissolved in anhydrous PhNO_2_ (ca. 1.00 mL/mmol eq) under an argon atmosphere (balloon) in a large two-necked flask fitted with a reflux condenser to give a beige suspension. The solution was chilled to 0 °C and allowed to stir, after which AlCl_3_ (4.00 eq) was added in three portions to give a clear brown solution. After 30 min, acyl chloride (1.10 eq) was added to the solution in a drop-wise fashion at ambient temperature. Once the addition was complete, the reaction was warmed to 65 °C and allowed to react for 21 h. Upon completion, the reaction was cooled to 0 °C. Ice-cold 3 M HCl (ca. 1.5 mL/mmol eq) was added to this cautiously, ensuring vigorous and continuous stirring to avoid exothermic run-off. Once quenched, the reaction mixture was extracted with Ethyl acetate (EtOAc) (3 × 30 mL), after which the volume of the combined organic phases was reduced by approximately half, and washed with 1 M NaOH (2 × 50 mL). The combined aqueous phase was acidified with 32% HCl to pH 1–2 and extracted again with EtOAc (3 × 50 mL). The organic phases were collected, washed with brine, dried over Na_2_SO_4_, and the solvent removed under vacuum. In the case of 3-phenyl-1-(2,4,6-trihydroxyphenyl)propan-1-one, the residue was further purified by trituration with DCM (5 × 10 mL). Crude material was carried through without further purification unless otherwise stated. The resulting crude oil was generally taken through to subsequent reactions without further purification. However, purification via Kugelrohr distillation gave the desired acylphloroglucinol as a yellow oil.

#### 2.1.2. General Synthesis of β-Triketones

Synthesis of the β-triketones was performed following the method of Perry et al. [38]. A two-necked round-bottomed flask was charged with anhydrous MeOH (ca. 5.00 mL/mmol eq) under an argon atmosphere, and cooled to 0 °C with stirring. Sodium metal (7.0 eq) was then added portion-wise. The reaction was allowed to reach room temperature and stirred until all of the sodium metal had reacted. A solution of acylphloroglucinol (1.0 eq) in anhydrous methanol (ca. 3.00 mL/mmol eq) was slowly added to the NaOMe solution. MeI (6.0 eq) was then added drop-wise, and the reaction warmed to reflux. After TLC indicated consumption of starting material (~3 h), the reaction was cooled to room temperature, and the solvent removed under vacuum. The residue was dissolved in minimal H_2_O, acidified with 3 M HCl, and extracted with EtOAc (3 × 20 mL). The combined organic phases were washed with brine (1 × 30 mL), dried over Na_2_SO_4_, and concentrated under vacuum. The β-triketones were subsequently purified by Kugelrohr distillation to give the desired compounds.

***Flavesone.*** Isolated as a yellow oil (0.871 g, 26% over 2 steps). ^1^H and ^13^C NMR data were found to be in agreement with the literature [38,39]. ^1^H NMR (CDCl_3_, 300 MHz): δ 18.46 (s, 1H), 3.80 (septet, 1H, *J* = 6.53 Hz), 1.45 (s, 6H), 1.37 (s, 6H), 1.18 (d, 6H, *J* = 6.78 Hz). ^13^C NMR (CDCl_3_, 125 MHz): δ 209.9, 208.6, 199.3, 196.9, 57.0, 52.2, 35.2, 24.3, 23.9, 19.1. GC-MS *m*/*z* (molecular ion, relative intensity): 252 (M^+^, 100%), 237 (M^+^-CH_3_, 39%), 209 [M^+^-CH(CH_3_)_2_, 29%].

***Leptospermone.*** Isolated as a yellow oil (1.98 g, 60% over 2 steps). ^1^H and ^13^C NMR data were found to be in agreement with the literature [38,39]. ^1^H NMR (CDCl_3_, 300 MHz): δ 18.39 (s, 1H), 2.90 (d, 2H, *J* = 6.93 Hz), 2.19 (s, 1H), 1.47 (s, 6H), 1.38 (s, 6H), 1.01 (d, 6H, *J* = 6.69 Hz). GC-MS *m*/*z* (molecular ion, relative intensity): 266 (M^+^, 100%), 251 (M^+^-CH_3_, 58%), 209 [M^+^-CH_2_CH(CH_3_)_2_, 36%].

***Isoleptospermone.*** Isolated as a yellow oil (0.627 g, 19% over 2 steps). ^1^H and ^13^C NMR data were found to be in agreement with the literature [38]. ^1^H NMR (CDCl_3_, 300 MHz): δ 18.45 (s, 1H), 3.62 (sextet, 1H, *J* = 6.77), 1.69–1.85 (m, 1H), 1.39–1.46 (m, 1H, hidden), 1.44 (d, 6H, *J* = 6.63 Hz), 1.36 (s, 6H, *J* = 6.81 Hz), 0.92 (t, 3H, *J* = 7.41 Hz). GC-MS *m*/*z* (molecular ion, relative intensity): 266 (M^+^, 93%), 251 (M^+^-CH_3_, 74%).

***Grandiflorone.*** Isolated as a yellow oil (1.05 g, 24% over 2 steps). ^1^H and ^13^C NMR data were found to be in agreement with the literature [38,40]. ^1^H NMR (CDCl_3_, 500 MHz): δ 18.26 (s, 1H), 7.18–7.30 (m, 5H), 3.34 (t, 2H, *J* = 7.70 Hz), 2.99 (t, 2H, *J* = 7.68 Hz). ^13^C NMR (CDCl_3_, 125 MHz): δ 210.0, 203.8, 198.8, 196.8, 140.7, 128.6, 128.6, 126.4, 109.4, 57.0, 52.0, 41.1, 31.0, 24.5, 24.0. GC-MS *m*/*z* (molecular ion, relative intensity): 314 (M^+^, 39%), 296 (M^+^-OH, 39%).

### 2.2. Drug Dilution Details for Antimicrobial Testing

Abametapir was obtained as a pure powder (Hatchtech, Pty Ltd., Melbourne, Australia) and dissolved in 100% dimethyl sulfoxide (DMSO; Sigma Aldrich, Castle Hill, New South Wales, Australia) to a stock concentration of 100 mM. This was then diluted in Mueller Hinton II Broth (MH II; Difco, Edwards Group Holding, Murarrie, Australia) in the first well of a 96-well plate (in triplicate; Greiner Bio-One, Interpath Services, Heidelberg, Victoria, Australia) to give a final starting concentration of 50 mM and 50% DMSO, in the first well.

Ivermectin was purchased as a 5 mg/mL (5.71 mM) Ivomec pour-on solution Vet N Pet Direct, Jimboomba, Australia), and diluted in MH II Broth to a starting concentration of 1.25 mg/mL (1.43 mM) (contains 20% isopropanol). The pour-on Ivomec solution is a commercial formulation prepared in isopropanol. This was used for assays involving Staphylococcus and Acinetobacter strains, but not Streptococcus strains, as the isopropanol in the pour-on solution interacted with the red blood cells in the sheep’s blood supplemented media. Ivermectin (pure, powdered) was purchased from Sigma Aldrich, and dissolved in 100% DMSO) to a stock concentration of 5 mg/mL (5.71 mM) and diluted in MH II Broth to a starting concentration of 2.5 mg/mL (2.86 mM).

Mānuka oil (KiwiHerb^®^) was obtained from Phil Rasmussen (Phytomed, New Zealand) and diluted in 100% DMSO to a stock concentration of 50% *v*/*v*. The starting concentration in the assay of mānuka oil, following dilution with MH II broth, was 25% *v*/*v*. The synthesized β–triketones (flavesone, isoleptospermone, leptospermone and grandiflorone) were diluted in 100% DMSO to a stock concentration of 200 mM, with a final starting concentration of 100 mM and 50% DMSO in the first well.

### 2.3. Bacterial Strains and Growth Conditions

*Acinetobacter baumannii* ATCC19606, ATCC17987, and BAA-1605 strains were obtained from Associate Professor Mark Blaskovich Institute for Molecular Biosciences, University of Queensland, Brisbane, Australia.

*Staphylococcus aureus* XEN29 was purchased from Caliper LifeSciences (Waltham, Massachusetts, United States) and *Staphylococcus aureus* CC75 strains M34 and M5 were provided by Dr. Deborah Holt at the Menzies School of Health Research, Charles Darwin University, Darwin [41]. These bacteria were cultured at 37 °C under aerobic conditions in Mueller Hinton II Broth (Difco, Edwards Group Holding, Murarrie, Queensland, Australia).

*Streptococcus pyogenes* strains 2967, 2031, and 8830 were obtained from Professor Sri Sriprakash (QIMR Berghofer MRI, Brisbane, Australia) and *Streptococcus dysgalactiae* subsp. *equisimilis* strains, MD10 and NS3396 were obtained from Professor David McMillan, (University of the Sunshine Coast, Sippy Downs, Australia). These bacteria were cultured at 37 °C in 5% CO_2_ in MH II Broth supplemented with 5% Sheep’s Blood (Thermo Fischer Scientific, Waltham, MA, USA). The strain information is summarized in Table 1.

The minimum inhibitory concentration (MIC) and minimum bactericidal concentration (MBC) were determined for the strains outlined in Table 1 using the broth microdilution method described in the protocol of the clinical and laboratory reference standards institute [42,43]. Two-fold serial dilutions of the test compounds were performed with MH-II media. Equal volumes of bacterial suspensions were added to each well (excluding sterility control wells) to give a final concentration of 5 × 10^5^ CFU/mL per well. Plates were incubated at 37 °C for 18 h in a 96-well flat bottom plate (Greiner Bio-One, Interpath Services, Heidelberg, Victoria, Australia ), under the conditions previously outlined. For all Streptococcus species MH II broth was supplemented with 5% Sheep’s blood. To validate the methodology, the antimicrobial effect of vehicles (DMSO and isopropanol) was tested using upper concentrations of 50% for each. Sterility control (i.e. sterile culture medium) and positive drug controls (ampicillin and ampicillin + sulbactam) were also tested (Appendix A). The MIC was determined after 24 h by visual inspection (i.e., recording the lowest concentrations required to elicit 100% inhibition of growth). Growth curves were also determined by OD_600_ measurements using a UV-Vis plate reader (BMG POLARstar Optima). Each assay was run with technical triplicates, and three biological replicates.

The minimum bactericidal concentration (MBC) was determined by subculturing wells with no visible growth onto agar plates (Acinetobacter and Staphylococcus MH II Agar and Streptococcus MH II Agar + 5% sheep’s blood) [44]. The plates were incubated at 37 °C at the previously outlined culture conditions and colony formation was examined after an overnight incubation. The MBC was defined as the lowest concentrations that exhibited no colony growth. A compound was deemed bactericidal if the MBC was ≤4 × MIC.

### 2.4. Statistical Analysis

Each assay was run with technical triplicates, and three biological replicates. The MIC and MBC values were expressed as statistical means (mM or % *v*/*v*) ± standard deviation (SD) and were determined using Graph Pad Prism 8. The MIC and MBC values for mānuka oil were expressed in % *v*/*v*. Tukey’s multiple comparisons test was used to assess the statistically significant difference (*p* < 0.05) between MIC and MBC values.

## 3. Results

The antibiotics used in these assays inhibited growth at the reported MIC and MBC values for the relevant bacteria (results in Appendix A). The MIC of DMSO was determined to be 12.5% *v*/*v* against all bacterial strains tested, and the MBC 25% *v*/*v*. At 12.5% DMSO the growth did not exceed the starting inoculum, except for *S. aureus*, where normal growth was observed. Except for *A. baumannii*, DMSO concentration was <12.5% in the respective MIC and MBC. For *A. baumannii*, MBC values of ≥25 mM, contain 12.5% DMSO which could have augmented the results.

### 3.1. Antibacterial Activity of Mānuka Oil and Its β-Triketones

Whole mānuka oil was effective at inhibiting the growth of all three *A. baumannii* strains at a concentration of ~3.125% (Table 2, Figure 2a). The MIC for the β-triketones against three *A. baumannii* strains were comparable to each other (MIC_flavesone_ = 6.25–13 mM; MIC_isoleptospermone_ = 13 mM; MIC_leptospermone_ = 13 mM; MIC_grandiflorone_ = 10–13 mM) with no significant difference in effectiveness between the four compounds (*p* > 0.05) (except for BAA1605–Flavesone) (Table 2, Figure 3a (representing type strains of the selected bacterial species)). Additionally, both mānuka oil and all the β-triketones showed bactericidal properties (i.e., MBC ≤ 4 × MIC). Mānuka oil was bactericidal at 3.1% for all three strains, whereas flavesone, isoleptospermone, leptospermone and grandiflorone showed MBC values of 25 mM, 21–25 mM, 21–25 mM, and 12.5–25 mM, respectively. There was no significant difference in MBC (*p* > 0.05) between the four individual β-triketones (except for BAA1605 –grandiflorone).

Against *S. aureus* strains, mānuka oil had both strong inhibitory and bactericidal properties (MIC = 0.30–0.39%; MBC = 0.39%). There is a significant difference between the MICs for all four β-triketones (*p* < 0.05), with the most effective individual β-triketone being grandiflorone (MIC_flavesone_ = 1.5–3.1 mM; MIC_isoleptospermone_ = 0.8–1.6 mM; MIC_leptospermone_ = 0.4–0.8 mM; MIC_grandiflorone_ = 0.13–0.16 mM) (Table 2, Figure 3a,b). For all three strains, there is a statistically significant difference in MIC values between grandiflorone (*p* < 0.05) when compared to flavesone, isoleptospermone and leptospermone, indicating that grandiflorone was the most effective triketone against *S. aureus*. Bacteriostatic activity was observed in all β-triketones against MRSA, and XEN NCTC8532.

Our results demonstrate that mānuka oil has both inhibitory and bactericidal properties against all three *S. pyogenes* strains, with MIC ranging between 0.2 and 0.41% for all three strains (Table 2). The individual β-triketones also exhibited both inhibitory and bactericidal properties against all three *S. pyogenes* strains, MIC_flavesone_ = 0.79–1.5 mM; MIC_isoleptospermone_ = 0.32–0.78 mM; MIC_leptospermone_ = 0.16–0.37 mM; MIC_grandiflorone_ = 0.086–0.16 mM) with grandiflorone showing the highest potency in all strains, and the flavesone the lowest (Table 2, Figure 3a). Grandiflorone also had the lowest MBC values compared to flavesone, leptospermone and isoleptospermone (Table 2, Figure 3b).

Against *S. dysgalactiae*, mānuka oil has both inhibitory and bactericidal properties with MICs between the two strains of 0.13–0.32% (Table 2, Figure 2a), and a mean MBC of ~0.65% (Table 2, Figure 2b). Additionally, the individual β-triketones demonstrated good inhibitory properties against both GGS strains (MIC_flavesone_ = 1.6–2.1 mM; MIC_leptospermone_ = 0.32–0.43 mM; MIC_isoleptospermone_ = 0.70–0.78 mM; MIC_grandiflorone_ = 0.14–0.17 mM). Grandiflorone and leptospermone were both more effective at inhibiting bacterial growth than flavesone (*p* < 0.05), and flavesone, leptospermone and isoleptospermone had higher MBC values than grandiflorone (MBC_flavesone_ = 10.4–12.5 mM; MBC_leptospermone_ = 6.25 mM; MBC_isoleptospermone_ = 4.17–8.30 mM; MBC_grandiflorone_ = 1.17–1.56 mM)**,** which performed significantly better (*p* > 0.05) (Table 2 Figure 3b).

Notably, both mānuka oil and the β-triketones showed between 2–140-fold greater potency (based on MIC values) against Gram-positive bacteria compared to the Gram-negative *A. baumannii* (Table 2).

### 3.2. Antibacterial Activity of Abametapir and Ivermectin

Unlike mānuka oil and the β-triketones, activity of abametapir against Gram-negative *A. baumannii* was found to be somewhat greater than in *Staphylococcus* sp. Similarly, ivermectin showed greater activity in *A. baumannii* than any of the Gram-positive species. The Ivomec pour-on solution was not suitable for the Streptococcus species, due to lysis of the red blood cells present in the culture media. Pure ivermectin was used instead for the experiments involving Streptococcus. Consequently, the results reported for these bacteria are not directly comparable to results reported for *A. baumannii* and *S. aureus*, as the pour-on solution utilized a different vehicle composition and may contain additives that affected compound-uptake and/or bacterial growth.

Abametapir was successful at inhibiting the growth of all three *A. baumannii* strains, with MIC values between 0.74–1.56 mM. This is significantly lower than the MIC for all four β-triketones (*p* < 0.05), indicating that abametapir is more effective at inhibiting the growth of Gram-negative organisms than flavesone, isoleptospermone, leptospermone and grandiflorone (Table 2, Figure 3a). Despite its inhibitory properties abametapir was not bactericidal against *A. baumannii*. Ivermectin (Ivomec) exhibited inhibitory activity against *A. baumannii*, with MIC values ranging from 0.21 to 0.39 mM, this is significantly lower than the MIC values for the β-triketones (*p* < 0.05). Ivermectin could not be classified as bactericidal against *A. baumannii* strains and MBC values could not be obtained within the concentration range tested.

Against *S. aureus*, abametapir showed moderate bactericidal activity (MIC = 1.3–1.56, MBC = 1.56–5.2) mM; Table 2), comparable to that observed for leptospermone and isoleptospermone. Moreover, this activity is significantly lower than that of flavesone (MBC) (*p* < 0.05), and significantly higher than the MBC observed for grandiflorone (*p* < 0.05). Additionally, ivermectin showed an MIC of 0.71 mM for all three strains. However, the MBC was higher than the concentration range tested.

The MIC of abametapir against all three strains of *S. pyogenes* (MIC = 2.8–3.1 mM) was comparatively higher than for *S. aureus* (MIC = 1.3–1.56 mM; Table 2). Additionally, abametapir is also observed to have significantly higher MICs than the β-triketones (*p* < 0.05). Despite the high similarity in MIC values across the strains, abametapir only exhibited bactericidal activity against *S. pyogenes* 2031 and 2967, but not 8830 (Table 2). The observed MIC for ivermectin against *S. pyogenes* strains 8830 and 2031 was determined to be 0.71 mM (in both cases), and 0.15 mM against 2967 (Table 2). This is significantly higher than the MIC of grandiflorone (*p* < 0.05), indicating that while ivermectin can inhibit the growth of *S. pyogenes*, the β-triketones appear to be more effective against Gram-positive organisms. Ivermectin exhibited no bactericidal activity against the three tested strains.

The MIC of abametapir against the two tested GGS strains was 3.1 mM (Table 2). This value is significantly higher than the MIC of the individual β-triketones (*p* < 0.05) against GGS. Additionally, abametapir had no bactericidal effect on GGS. A similar result was observed for ivermectin, where the MIC for the two strains was 0.58–0.71 mM (Table 2), with no observed bactericidal effect.

## 4. Discussion

Scabies is strongly correlated with secondary bacterial infections that can lead to severe downstream health consequences. This is especially well documented in lower socio-economic countries with poor resources and overcrowded living conditions. Damage to the host’s skin from the burrowing action of the parasite and the scratching due to extreme itch, contribute to the disruption of the skin barrier. This disruption leads to secondary bacterial infections, particularly with *S. pyogenes* and *S. aureus*. Both of these pathogens are opportunistic bacteria that have their own arsenal of molecules capable of subverting the hosts complement system [45,46,47]. This combined with the immunosuppressive molecules excreted by the scabies mites leave the host susceptible to severe secondary bacterial infections, such as rheumatic fever and rheumatic heart disease, both of which are highly prevalent in areas where scabies is endemic [10]. This link between scabies and severe downstream bacterial disease necessitates new research into treatment options that target both the mite and the opportunistic bacteria. Here, we investigate potential antimicrobial activities of current and emerging scabicides against clinical isolates of bacterial pathogens commonly associated with scabies infections to investigate whether scabicides may offer both anti-scabies and anti-bacterial protection. Preliminary data indicate that these effects may be elicited below their respective scabicidal concentrations, suggesting they could act as potent dual therapeutic agents.

Mānuka oil (and to a lesser extent, its naturally occurring β-triketones) has been reported to have bacteriostatic and bactericidal properties against numerous Gram-positive and Gram-negative bacterial pathogens, and its activity against *S. aureus* is well characterized [23]. While mānuka oil has previously been found to be active against other Gram-negative pathogens such as *E. coli* and *P. aeruginosa*, activity is generally found to be low [25]. This is reflected in trends in Gram positive and negative antimicrobial activity seen here, where activity against *A. baumannii* strains was significantly reduced compared to *S. aureus* and Streptococcus species. Lower activity against *A. baumannii* by mānuka oil is not surprising considering that tea tree oil, another EO with a high β-triketone content, has also shown bactericidal activity against *A. baumannii* 19606 [48].

Our research demonstrated that the four most prevalent β-triketones present in mānuka oil have anti-bacterial activity against key bacterial species implicated in scabies-associated secondary infection. Additionally, it is of note that grandiflorone appears to be moderately more effective compared to the other β-triketones. This is in-line with findings from van Klink et al. who were the first to describe the anti-bacterial properties of the individual β-triketones, and found that grandiflorone was the most effective natural β-triketone against *S*. *aureus* [30]. It has been noted that, when grown in the presence of bacteria, the mānuka plant increases the concentration of grandiflorone present in the leaves [49]. Similar to mānuka oil, the β-triketones were less active against the Gram-negative bacteria tested here. This is likely due to the differing structure of the cell wall, as Gram-negative bacteria have a more complex cell wall comprising an outer membrane that lies outside the comparatively thinner peptidoglycan layer [50]. The activity of essential oils has been attributed to its hydrophobicity, which increases membrane permeability and can lead to leakage of cell contents, reduction in proton motive force and decreased ATP synthesis, all of which are critical for bacterial survival [50,51]. This effect is typically less pronounced in Gram-negative bacteria as the cell wall does not as readily allow entrance of hydrophobic molecules. This could contribute to the results reported here with the MIC and MBC values for *A. baumannii*, being significantly higher than those for the Gram-positive bacteria tested. Despite this reduced activity, mānuka oil and its β-triketones do have antibacterial activity against the tested scabies-associated bacteria. This combined with its scabicidal properties, make mānuka oil an ideal drug candidate for the treatment of scabies.

In 2017, scabies was recognized as a neglected tropical disease, this declaration came with a call to develop novel therapeutics. Since this declaration there have been few promising candidate drugs proposed. One of these is the novel compound abametapir, a metalloproteinase inhibitor, and new-generation lousicide approved by the FDA for use on human skin at a concentration ~40 mM. Due to its mechanism of action, and the role of bacterial extracellular metalloproteases (BEMPs) in degrading environmental proteins and peptides for heterotrophic nutrition, we wanted to understand whether this emerging scabicide also has anti-bacterial activity [52]. Bacterial proteases are essential in cell viability, stress response and pathogenicity, and have been highlighted as an ideal antimicrobial target in the current age of emerging drug resistance [53]. The aim of this study was to determine whether abametapir has anti-microbial properties against scabies-associated bacteria, to better understand the value of this emerging scabicide.

Abametapir exhibited inhibitory activity against the Gram-positive and Gram-negative bacteria tested here. Comparatively though, it was only bactericidal to *S. aureus*. The reason for this difference could be due to differing metalloprotease functions. *S. aureus* has a well characterized metalloprotease, aureolysin, which has many important biological functions that contribute to its pathogenicity, and has been demonstrated to be important for bacterial survival in human whole blood [54]. Aureolysin is important for nutrition acquisition and evasion of the hosts complement system. An inhibition of aureolysin by abametapir could contribute to the anti-microbial properties observed against *S. aureus* [55]. Comparatively, the role of *S. pyogenes* and *S. dysgalactiae* metalloproteases is not reported to be associated with virulence, possibly due to Streptococcus sp. and particularly *S. pyogenes* relying primarily on cysteine proteases for pathogenicity and virulence [56]. However, both of these Streptococcus species have methionine aminopeptidase, a metalloprotease that is essential for cleaving N-terminal methionine from nascent proteins [57]. Studies have demonstrated that methionine is essential for the survival of Group B Streptococcus (GBS), and is an essential component for nutrient uptake. It is therefore likely that this could be a potential target of abametapir in Streptococcus species [57]. Abametapir was effective at inhibiting the growth of *A. baumannii*, however, exhibited no bactericidal activity. *A. baumannii* is a clinically important nosocomial pathogen, due to its exceptional virulence and drug resistance. *A. baumannii* has a large arsenal of virulence factors that enable it to subvert the host immune response and are essential for nutrient acquisition. Amongst these virulence factors are metalloproteases that are important in type II secretion systems [58]. *A. baumannii* also utilizes an ATP-dependent integral membrane metalloprotease FtsH, which is essential for protein regulation. Disruption of FtsH results in growth arrest and cell division defects [59,60]. This metalloprotease could be a potential target of abametapir in *A. baumannii*, and could explain the inhibitory, but not bactericidal activity of this drug. FtsH is ubiquitous to all bacteria [59], which could indicate that abametapir may have broad-spectrum antimicrobial activity, an interesting property for a candidate scabicide. Further research is required to understand the molecular target of abametapir in these species.

The currently frequently used scabicide ivermectin is a broad-spectrum anti-parasitic macrocyclic lactone that induces hyperpolarization of cells through acting on glutamate-gated chloride channels [61]. There is emerging in vitro evidence that ivermectin has anti-bacterial effects. This was first described in Mycobacterium species [62]; however, in relation to its role as a scabicide there is now emerging in vitro evidence of anti-bacterial activity against some *S. aureus* strains [61]. However, there is evidence that this is strain specific, with resistance being reported in ~90% of tested strains [61]. It is believed that this resistance is due to the overexpression of efflux pumps [63]. Anti-bacterial activity of ivermectin has been anecdotally observed in a mass-drug administration (MDA) study performed by Romani et al. [64] in the Solomon Islands. This MDA was a small community- based trial aimed at decreasing the prevalence of scabies and secondary impetigo, through the administration of ivermectin alone, or ivermectin and azithromycin [64]. The authors reported a significant decrease in the prevalence of Impetigo (74%) after treatment, and no significant difference in impetigo rates between the ivermectin-treated group and the combined antibiotic treatment group [64]. This could be a result of both a reduction in the scabies prevalence, and possibly some antibacterial activity of ivermectin. In our study, and keeping in mind that a different vehicle had to be used for testing *S. pyogenes*, we observed that ivermectin did exhibit inhibitory properties against the Gram-positive bacteria *S. aureus*, *S. pyogenes*, and *S. dysgalactiae* subsp. *equisimilis* at a similar concentration. However, we observed no bactericidal activity. Additionally, we noted that ivermectin exhibited inhibitory properties against Gram-negative *A. baumannii*. Once again, we did not observe bactericidal activity. Antibiotics that contain a macrocyclic lactone ring typically inhibit bacterial protein biosynthesis, and this is the likely mechanism of action of ivermectin [65]. Further research is required to determine how ivermectin may be effective at treating both scabies, and scabies-related impetigo in patients.

## 5. Conclusions

The increasing global burden of scabies necessitates the development of new single-dose treatments, and with the call for novel scabicides we propose that it is similarly important to consider the antimicrobial activity of these compounds due to the strong correlation between scabies and severe secondary bacterial infections. Here we have demonstrated that two novel scabicides abametapir, and mānuka oil, as well as its β-triketones, exhibit antibacterial properties against both Gram-positive and Gram-negative pathogens of concern in the scabies infection. The demonstrated MIC and MBC values for abametapir are higher than what would be clinically acceptable for an individual antimicrobial agent, however, the MIC and MBC values are significantly below the concentration required for effective scabies treatment (~40 mM, and is the FDA approved dosage). The effectiveness of both of these compounds makes them highly interesting scabicide candidates for future drug discovery research. However, the mānuka oil components leptospermone and grandiflorone may be interesting to follow up as potential antimicrobials due to their significantly lower MIC values. It is also of note that the MIC values for these β-triketones are similar (maximum two-fold difference) across the strains tested, regardless of drug resistance phenotype, indicating the non-specific mode of action against these bacteria. The finding that ivermectin offers some inhibitory activity against several pathogens of concern, including *S. pyogenes*, is in line with findings from MDA studies that demonstrated a decrease in impetigo in ivermectin-treated populations, indicating that ivermectin could be considered as a first-line drug when treating scabies-related impetigo. Notably, the MIC values of ivermectin are much higher than would be acceptable for a clinical antimicrobial, similar to abametapir.

## Figures and Tables

**Figure 1 biomedicines-10-03287-f001:**
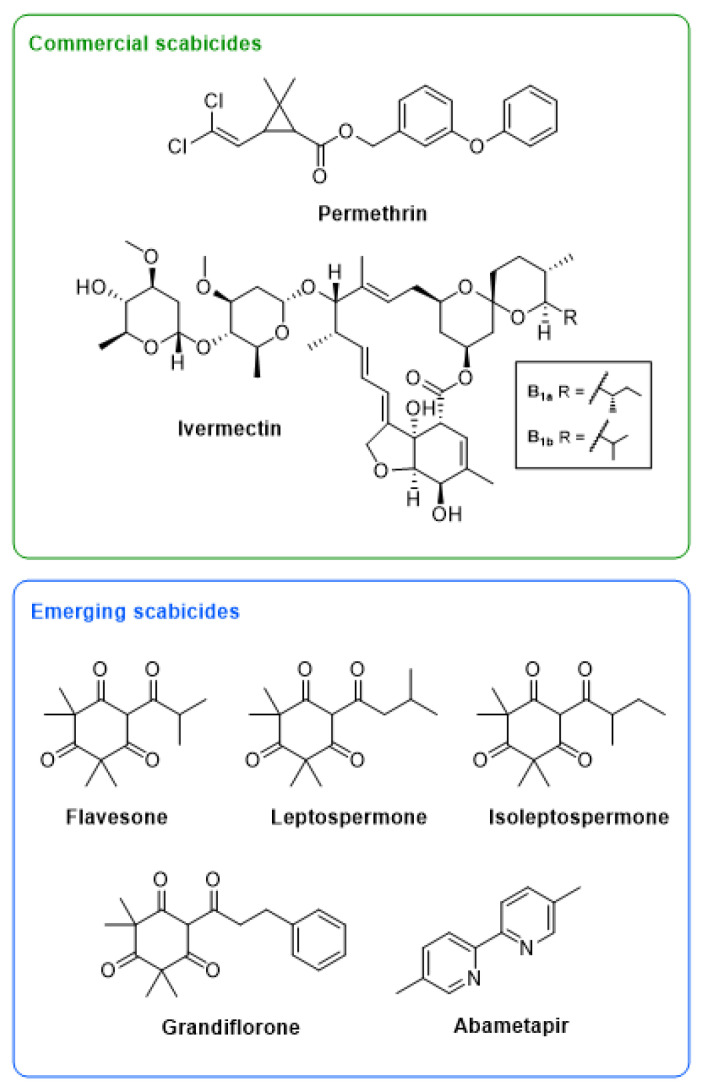
Structures of two commercial scabicides, ivermectin and permethrin, and five emerging scabicides, including a commercial head lice drug, abametapir, and four β-triketones commonly found in mānuka oil, flavesone, leptospermone, isoleptospermone, and grandiflorone.

**Figure 2 biomedicines-10-03287-f002:**
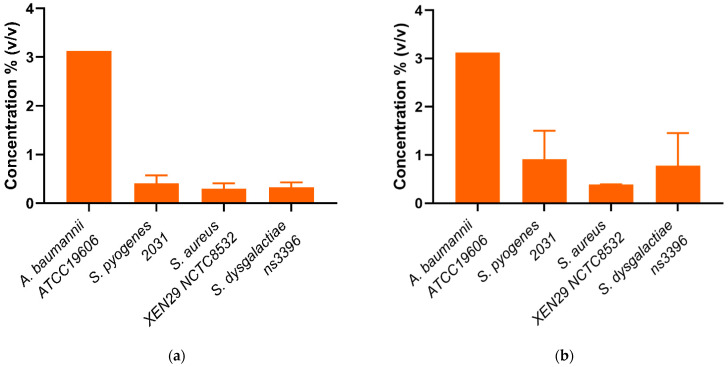
(**a**) Minimum inhibitory concentrations (MIC) and (**b**) Minimum bactericidal concentrations (MBC) of mānuka oil (% *v*/*v*) against four scabies-associated bacteria.

**Figure 3 biomedicines-10-03287-f003:**
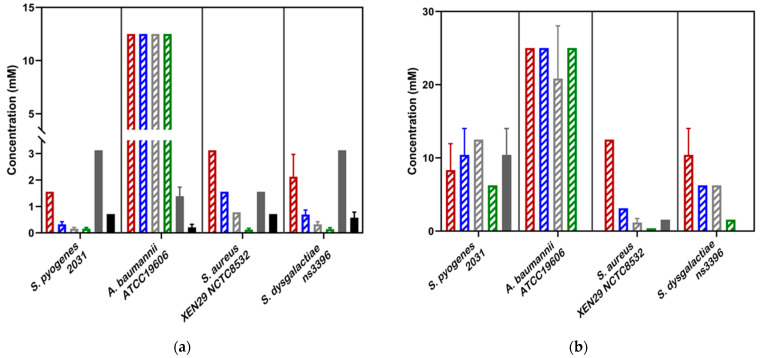
(**a**) Minimum inhibitory concentrations (MIC ) and (**b**) Minimum bactericidal concentrations (MBC) of novel scabicides against four scabies-associated bacteria. For all Streptococcus strains, the pure Ivermectin formulation was used. For *S. aureus* and *A. baumannii*, the Ivomec Pour-on Solution was used. Ivermectin did not show bactericidal activity against all strains tested. In the above bar chart, the β-triketones are shown in a striped pattern, with Flavesone in red, isoleptospermone in blue, leptospermone in grey and grandiflorone in green stripes. Abemetapir is presented in solid grey and ivermectin in in solid black.

**Table 1 biomedicines-10-03287-t001:** Bacterial strains used in this study.

**Bacteria (Gram Positive)**	**Characteristics**
*S. aureus XEN29*	NCTC8532, *Kan^R^*
*S. aureus CC75, M34*	Pyoderma clinical isolate, MRSA, lacks staphyloxanthin
*S. aureus CC75, M5*	Pyoderma clinical isolate, MSSA, lacks staphyloxanthin
*S. pyogenes 2967*	emm1, M1, PSGN
*S. pyogenes 2031*	emm1, M1, type strain
*S. pyogenes 8830*	Pyoderma clinical isolate, emm97
*S. dysgalactiae subs. equisimilis MD10*	STG 6
*S. dysgalactiae subs. equisimilis NS3396*	Clinical isolate, Acute Rheumatic Fever patient, STG 480
**Bacteria (Gram negative)**	**Characteristics**
*A. baumannii ATCC19606*	Biofilm forming strain, type strain (urine)
*A. baumannii ATCC17987*	Drug sensitive (fatal meningitis)
*A. baumannii BAA-1605*	Clinical isolate, sputum, multi-drug resistant

Determining the Minimum Inhibitory Concentration (MIC) and Minimum Bactericidal Concentration (MBC).

**Table 2 biomedicines-10-03287-t002:** MIC and MBC values for commercial mānuka oil, flavesone, leptospermone, isoleptospermone, grandiflorone, abametapir, ivermectin (topical formulation) and pure ivermectin against pathogenic bacterial strains, *Acinetobacter baumannii*, *Staphylococcus aureus*, *Streptococcus pyogenes* and *Streptococcus dysgalactiae*.

	*A.baumannii ATCC19606*		*A.baumannii ATCC17978*		*A.baumannii BAA1605*		*S. aureus XEN29 NCTC8532*		*S. aureus MSSA CC75 M5*		*S. aureus MRSA CC75 M34*		*S. pyogenes 2031*		*S. pyogenes 2967*		*S. pyogenes 8830*		*S. dysgalactiae MD10*		*S. dysgalactiae ns3396*	
	MIC (mM)	MBC (mM)	MIC (mM)	MBC (mM)	MIC (mM)	MBC (mM)	MIC (mM)	MBC (mM)	MIC (mM)	MBC (mM)	MIC (mM)	MBC (mM)	MIC (mM)	MBC (mM)	MIC (mM)	MBC (mM)	MIC (mM)	MBC (mM)	MIC (mM)	MBC (mM)	MIC (mM)	MBC (mM)
Flavesone	12.5 ± 0	25 ± 0	10 ± 3	25 ± 0	6.25 ± 0	25 ± 0	3.125 ± 0	12.5 ± 0	1.56 ± 0	12.5 ± 0	1.9 ± 0.7	12.5 ± 0	1.56 ± 0	8.3 ± 3.1	1.0 ± 0.4	1.91 ± 0.7	0.78 ± 0	4.7 ± 2.3	1.56 ± 0	10.4 ± 3.6	2.08 ± 0.78	12.5 ± 0
Isoleptospermone	12.5 ± 0	25 ± 0	12.5 ± 0	21 ± 7	12.5 ± 0	25 ± 0	1.56 ± 0	3.1 ± 0	0.78 ± 0	5.2 ± 1.6	1.1 ± 0.4	6.25 ± 0	0.32 ± 0.10	10.4 ± 3.1	0.78 ± 0	1.17 ± 0.58	0.39 ± 0	3.125 ± 0	0.78 ± 0	4.17 ± 1.8	0.70 ± 0.17	8.3 ± 3.6
Leptospermone	12.5 ± 0	21 ± 7	12.5 ± 0	21 ± 7	12.5 ± 0	25 ± 0	0.78 ± 0	1.2 ± 0.4	0.35 ± 0.09	2.6 ± 0.8	0.65 ± 0.19	3.1 ± 0	0.16 ± 0.05	12.5 ± 0	0.37 ± 0.06	1.17 ± 0.43	0.24 ± 0.09	6.25 ± 0	0.43 ± 0.13	6.25 ± 0	0.32 ± 0.10	6.25 ± 0
Grandiflorone	12.5 ± 0	25 ± 0	10 ± 0.3	25 ± 0	12.5 ± 0	12.5 ± 0	0.12 ± 0.05	0.39 ± 0	0.15 ± 0.05	1.3 ± 0.4	0.16 ± 0.05	0.52 ± 0.19	0.16 ± 0.04	6.25 ± 0	0.13 ± 0.04	0.195 ± 0	0.086 ± 0.060	0.780 ± 0	0.17 ± 0.04	1.17 ± 0.78	0.14 ± 0.05	1.56 ± 0.0
Mānuka Oil ^[a]^	3.1 ± 0	3.1 ± 0	3.1 ± 0	3.1 ± 0	2.6 ± 0.8	3.1 ± 0	0.30 ± 0.101	0.39 ± 0	0.32 ± 0.10	0.39 ± 0	0.39 ± 0	0.39 ± 0	0.41 ± 0.16	0.91 ± 0.60	0.28 ± 0.13	0.45 ± 0.30	0.20 ± 0.11	0.78 ± 0.68	0.13 ± 0.05	0.52 ± 0.23	0.32 ± 0.10	0.78 ± 0.68
Abametapir	1.4 ± 0.3	-	0.74 ± 0.13	-	1.56 ± 0	-	1.56 ± 0	1.56 ± 0	1.3 ± 0.4	5.2 ± 1.6	1.3 ± 0.4	5.2 ± 1.6	3.12 ± 0	10.42 ± 3.1	3.12 ± 0	10.42 ± 3.1	2.8 ± 0.5	25 ± 0	3.12 ± 0	-	3.12 ± 0	-
Ivomec (formulation)	0.21 ± 0.12	-	0.23 ± 0.13	-	0.39 ± 0.28	-	0.71 ± 0	-	0.71 ± 0	-	0.71 ± 0	-	n/a	n/a	n/a	n/a	n/a	n/a	n/a	n/a	n/a	n/a
Ivermectin	n/a	n/a	n/a	n/a	n/a	n/a	n/a	n/a	n/a	n/a	n/a	n/a	0.71 ± 0	-	0.15 ± 0.04	-	0.71 ± 0	-	0.71 ± 0	-	0.58 ± 0.21	-
DMSO ^[a]^	12.5 ± 0	25 ± 0	12.5 ± 0	25 ± 0	12.5 ± 0	25 ± 0	12.5 ± 0	25 ± 0	12.5 ± 0	25 ± 0	12.5 ± 0	25 ± 0	12.5 ± 0	25 ± 0	12.5 ± 0	25 ± 0	12.5 ± 0	25 ± 0	12.5 ± 0	25 ± 0	12.5 ± 0	25 ± 0
Isopropanol ^[a]^	6.25 ± 0	12.5 ± 0	6.25 ± 0	12.5 ± 0	6.25 ± 0	12.5 ± 0	12.5 ± 0	25 ± 0	12.5 ± 0	25 ± 0	12.5 ± 0	25 ± 0	n/a	n/a	n/a	n/a	n/a	n/a	n/a	n/a	n/a	n/a

^[a]^ Mānuka oil, DMSO, Isopropanol MIC and MBC values are given as *v*/*v*%. - indicates that there was no bactericidal activity observed. n/a—not tested.

## Data Availability

All the data is contained within the manuscript and Appendix A.

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
