# Peer review of "Investigating the Antibacterial Properties of Prospective Scabicides"

_biomedicines, 2022, doi:10.3390/biomedicines10123287_

Round 1

Reviewer 1 Report

Dear Authors,

you paper is of great interest, especially because in this period in many countries we are facing with resistence or pseudoresistence to permethrin. The content of the manuscript is of good quality and the results are well presented.

I humbly ask you to modify the first sentence of your manuscript "is amongst the most common dermatological skin diseases worldwide". It's true that the incidence is raising but in European countries cannot be considered among the most common skin diseases.

Moreover, the superinfections in scabies are uncommon in European countries; this should be stated in the manuscript (Esposito L, Veraldi S. Skin bacterial colonizations and superinfections in immunocompetent patients with scabies. Int J Dermatol. 2018).

Kind regards

Author Response

your paper is of great interest, especially because in this period in many countries we are facing with resistence or pseudoresistence to permethrin. The content of the manuscript is of good quality and the results are well presented.

RESPONSE: Thank you.

Reviewer 2 Report

The manuscript by Fischer etc. reported the anti-bacterial properties of two novel scabicides Abametapir and Mānuka Oil against a few common scabies-associated bacteria. While it might be helpful to the community to have better treatment options of scabies, and the manuscript was well written and organized, the reviewer, however, does not suggest acceptance of the manuscript in Biomedicines with the data presented at this point.

The data are essentially just the phenotypic testing against a few bacteria pathogens, further studies are recommended to conduct, for example, maybe in a mouse model. 

Author Response

I humbly ask you to modify the first sentence of your manuscript "is amongst the most common dermatological skin diseases worldwide". It's true that the incidence is raising but in European countries cannot be considered among the most common skin diseases.

RESPONSE: Thank you that is a good point. The sentence has been modified accordingly. The beginning of the abstract now reads: “Scabies is a dermatological disease found worldwide. Mainly in tropical regions, it is also the cause of significant morbidity and mortality due to its association with potentially severe secondary bacterial infections.”

Round 2

Reviewer 1 Report

No comments

Reviewer 2 Report

The authors did not make any response to my previous comments, maybe due to that they did not see the report. From the perspective that this work has employed scabicides to test the antimicrobial properties against common scabies-associated bacteria, and the MIC and MBC values are significantly below the concentration required for effective scabies treatment, this work is interesting to the community.